# SARS-CoV-2 Delta Variant: Interplay between Individual Mutations and Their Allosteric Synergy

**DOI:** 10.3390/biom12121742

**Published:** 2022-11-23

**Authors:** Kevin C. Chan, Yi Song, Zheng Xu, Chun Shang, Ruhong Zhou

**Affiliations:** 1Institute of Quantitative Biology, College of Life Sciences, Zhejiang University, Hangzhou 310058, China; 2Shanghai Institute for Advanced Study, Zhejiang University, 799 Dangui Road, Shanghai 201203, China; 3BirenTech Research, Shanghai 201112, China; 4Department of Chemistry, Columbia University, New York, NY 10027, USA; 5The First Affiliated Hospital, School of Medicine, Zhejiang University, Hangzhou 310058, China

**Keywords:** SARS-CoV-2, SARS-CoV-2 variants, molecular dynamics simulations, binding free energy, free energy perturbation, allostery, protein-protein interactions

## Abstract

Since its first appearance in April 2021, B.1.617.2, also termed variant Delta, catalyzed one major worldwide wave dominating the second year of coronavirus disease 2019 (COVID-19) pandemic. Despite its quick disappearance worldwide, the strong virulence caused by a few point mutations remains an unsolved problem largely. Along with the other two sublineages, the Delta variant harbors an accumulation of Spike protein mutations, including the previously identified L452R, E484Q, and the newly emerged T478K on its receptor binding domain (RBD). We used molecular dynamics (MD) simulations, in combination with free energy perturbation (FEP) calculations, to examine the effects of two combinative mutation sets, L452R + E484Q and L452R + T478K. Our dynamic trajectories reveal an enhancement in binding affinity between mutated RBD and the common receptor protein angiotensin converting enzyme 2 (ACE2) through a net increase in the buried molecular surface area of the binary complex. This enhanced binding, mediated through Gln493, sets the same stage for all three sublineages due to the presence of L452R mutation. The other mutation component, E484Q or T478K, was found to impact the RBD-ACE2 binding and help the variant to evade several monoclonal antibodies (mAbs) in a distinct manner. Especially for L452R + T478K, synergies between mutations are mediated through a complex residual and water interaction network and further enhance its binding to ACE2. Taking together, this study demonstrates that new variants of SARS-CoV-2 accomplish both “attack” (infection) and “defense” (antibody neutralization escape) with the same “polished sword” (mutated Spike RBD).

## 1. Introduction

Before its replacement by the Omicron variant in late 2021, the severe acute respiratory syndrome coronavirus 2 (SARS-CoV-2) Delta variant constituted about two third of the sequenced viruses across the globe. While a widely spreading strain may not necessarily imply a significant jump in binding affinity to human receptor ACE2 (hACE2), as well illustrated for the overspreading Omicron variant which binds to hACE2 at a similar affinity compared to previous strains, the Delta antigenic drift marked the ultimate optimization of hACE2 bindings to achieve a high transmissibility of SARS-CoV-2 [1,2,3,4,5]. The early appearance of Spike mutations, such as D614G in B.1 lineage, led to just 20% increased infectiousness but a rapidly growing global dominant strain in the initial period of the COVID-19 pandemic [6]. The next jump in infectiousness appeared in late 2020 also originated from Spike mutations, including N501Y in B.1.1.7 lineage, termed the Alpha variant, which had an approximate 50% increased transmissibility beyond that of D614G [7]. The Delta variant was reported to be three-fold more transmissible compared to the very original strain [8] and 60% more transmissible than the Alpha variant [5]. These numbers highlight the serious need for the studies of related mutations and especially the underlying molecular mechanisms of how individual mutations or epistasis in SARS-CoV-2 would affect its bindings to hACE2.

The ancestral lineage of the Delta variant–B.1.617 was originally identified in October 2020 in India and has since dominated the sequenced viruses circulating in many countries [9]. The lineage was subdivided into three sublineages, including the Delta variant (B.1.617.2) as a variant of concern (VOC), and two more subtypes, the Kappa variant (B.1.617.1) and B.1.617.3, as variants of interest (VOIs). These sublineages harbor diverse mutations in the receptor-binding motif (RBM) of Spike RBD (receptor binding domain). Some of these mutations were identified in pre-existing strains. For example, the L452R mutation appears in all the three subtypes and is the defining mutation of another VOC, the Epsilon variant (B.1.429), which emerged late 2020 in California of United States. An increase in transmissibility of about 20% was reported for lineages B.1.427/B.1.429 due to the L452R substitution [10,11]. Another mutation E484Q, presented in both subtypes B.1.617.1 and B.1.617.3, was suggested to be functionally similar to a highly antibody-evasive mutation E484K found in VOCs Beta and Gamma variants (B.1.351 and P.1) [12,13]. Interestingly, this mutation (E484Q) has likely reverted in the Delta sublineage as opposed to its presence in the ancestral lineage. Instead, the Delta variant harbors a rather unique T478K mutation. To date, this T478K mutation has been found in almost all widely spreading Omicron subtypes, indicating a clear acquisition of the mutation against evolutionary pressure. However, how the emergence of a single T478K mutation would affect viral replication and/or transmissibility remains unknown.

While natural infection, vaccination and cocktail of mAbs has shown to well handle the early emergence of SARS-CoV-2 variants with single mutation [14,15,16,17], emergence of variants, especially the Omicron variants, accumulating a large number of mutations in different Spike domains poses a more severe challenge dealing with divergent strains able to evade polyclonal responses [18,19]. Recently, it was confirmed that the escape of variants to neutralization were due to a reduction or loss of binding of neutralizing monoclonal antibodies (mAbs) to infected cells [5,8,20,21,22,23]. As a number of crystallography or cryogenic electron microscopy (cryo-EM) structures of Spike variants in complex with mAbs became available [1,2,24,25,26,27,28,29], quantitative measurements, either experimental or theoretical, on the binding affinities between mAbs and Spike domains were highly demanded. MD simulations, as a generally accepted computational tool, often complement experiments in investigating the detailed molecular interactions at the interfaces of macro-biomolecules and exhibit great potential in bridging various structural characterizations of RBD-antibody complexes with the level of immunoevasion from biochemical assays [30,31]. So far, there have been a few computational studies on the binding affinities between Spike RBD variants and biomolecules including human ACE2 receptor domain and mAbs. While some early studies focused primarily on single mutation variants [32,33,34,35,36,37], many follow-up studies discussed about multiple mutations [38,39,40,41,42,43,44,45,46,47,48]. These studies employed various free energy calculations, mostly through end-state methods, to quantify the effect of mutations to molecular interactions between biomolecules. Single mutations such as E484K, N501Y and L452R were reported to both increase RBD-ACE2 affinity and facilitate antibody evasion [32,34,35], other mutations such as T478K were only preliminarily suggested to induce antibody escape because of its presence near the epitope region of some mAbs. The fact that some trending variants involve mutations that are far away from molecular interaction interfaces has led to the notion of allostery (i.e., long-range synergetic effect) between mutations in some recent computational studies [44,46].

In this work, we presented a computational study of the SARS-CoV-2 Spike RBD, concerning especially mutations that define sublineages of B.1.617 (L452R, T478K and E484Q), including the Delta variant. The Spike RBD was investigated in complex with the human ACE2 receptor domain as well as some featured mAbs. By means of atomistic MD simulations, we revealed dynamic interactions behind the enhanced RBD-ACE2 bindings that harbor combinative mutation sets, L452R + E484Q and L452R + T478K. We further provided molecular details through well-established correlation analysis and a novel water-mediated network analysis as the mechanistic supplement to the observed synergetic effect in such enhanced binding. By validating our comprehensive set of alchemical binding free energy (BFE) calculations against both previous experimental [5,28,29,49] and computational [33,50,51] works on these mutation sets, we revealed the complex interplay between individual mutations at distance and quantitatively estimate the level of evasion for mAbs.

## 2. Materials and Methods

Both MD and FEP simulations were performed using CHARMM forcefields [52,53]. The simulations were initiated from the crystal structures from Protein Data Bank (PDB) of RBD-ACE2 complex (PDB ID: 6M0J), RBD-B38 complex (PDB ID: 7BZ5), RBD--COVA2-39 complex (PDB ID: 7JMP), RBD--P2B-2F6 complex (PDB ID: 7BWJ), and RBD-CR3022 complex ((PDB ID: 6YLA). All systems were then solvated with TIP3P water molecules [54] and 0.15 M NaCl. All simulations were performed on GPUs using Gromacs [55] for MD simulations and NAMD [56] for FEP simulations (see the Appendix A for details). The dual topology was implemented using VMD [57]. FEP calculations were initiated from uncorrelated snapshots of independent MD simulations. Each system was carefully mutated using λ increments of maximum 0.04, totaling at least 21 FEP windows each of at least 600 ps (see the Appendix A for details). Error bars for BFE were standard errors. The buried areas, hydrogen bonds, and salt bridges were computed using VMD [57]. For the FEP simulations, water-mediated interactions were computed for only the frames from the last five alchemical states, representing the mutated states (see the Appendix A for details).

## 3. Results

### 3.1. MD Simulations Revealed a Tighter Binding to ACE2 by Mutated RBDs

To elucidate the effects of mutations found in the SARS-CoV-2 B.1.617 lineage, including the Kappa (κ, B.1.617.1) and the Delta (δ, B.1.617.2) variants, we first performed unbiased MD simulations of binary complexes between the hACE2 domain and the spike RBD containing two sets of double mutations (L452R + E484Q as RBD_mut1_ and L452R + T478K as RBD_mut2_). The complexes were solvated with explicit water molecules and neutralized with NaCl ions at a physiological concentration (Appendix A). In several previous studies [58], a native RBD was reported to form stable complex with ACE2 showing minor structural deviations. Similarly in our simulations, both sets of mutations have shown very limited effects on the overall complex structure from two trial runs for each variant (with each lasting 1 μs), indicating a stable binding between the mutated RBD and the ACE2 domain (Figure 1A). In addition, the residues on the RBD in contact with the ACE2 domain (termed the native contacts) were largely retained for both mutated complexes (Figure 1B), preserving primarily hydrophobic regions (Appendix A) and hydrogen-bonding networks [58] as a highly optimized collection of interactions found in the SARS-CoV.

Given the fact that both sets of mutations would enhance infectiousness of the virus [10], whether the mutated residues had facilitated the molecular recognition of ACE2 through the RBD could be inferred from direct investigation of the binding region. Intriguingly, despite the highly conserved interaction interface, we observed an increase in buried areas between the mutated RBD and ACE2. Compared to a buried area of 853 ± 62 Å^2^ for the wild-type RBD-ACE2 complex (851 Å^2^ for the crystal structure), the area for RBD_mut1_ and RBD_mut2_ are 908 ± 54 and 916 ± 50 Å^2^ respectively in our MD simulations (Figure 2A). Such an increase in the protein-protein contact area is highly consistent with cryo-EM structures [29] and often indicates a tighter binding, leading to enhanced infectiousness in viral activities [59].

Consistent with the shape complementarity and structural details, the observed higher contact (or tighter binding) may be attributed from bringing residue Gln493 found in previously defined contact region 2 (CR2) [58] of RBD closer to the N-terminal helix of ACE2 (Figure 2B). In our MD simulations of both RBD_mut1_ and RBD_mut2_ in complex with ACE2, residue Glu35 on the N-terminal helix of ACE2 showed a slight increase in hydrogen-bond forming with Gln493 from the mutant RBD compared to the wild-type (WT) (Appendix A), in agreement with the tighter binding for mutants. Besides Glu35, residue Lys31 on the N-terminal helix also remotely formed hydrogen bonds with the sidechain of Gln493 of RBD in all our simulations. It is worth noting that, although Glu484 of RBD contributes an electrostatic complementarity through occasionally forming salt-bridges with the Lys31 in the region (Appendix A), we did not observe a direct hydrogen bond network between Glu484 and any residues of ACE2 in our simulations. More interestingly, the point mutation site L452R, a common mutation in all the three sublineages of B.1.617, locates right underneath Gln493. We therefore anticipate that the L452R mutation increased the structural rigidity around Gln493 and further altered the geometrical surface of RBD for better shape complementarity to ACE2. In accordance, a significant decrease in spatial distance between the sidechain of Gln493 and the N-terminal helix of ACE2 was observed, which is also in good correlation with the increase in the buried areas between the proteins (Appendix A). Highlighting the role of Gln493 at mutated binary interfaces also coincides with the one-carbon extension of the sidechain of residue 493 when evolving from asparagine (Asn) in SARS-CoV to glutamine (Gln) in SARS-CoV-2. This Asn-to-Gln mutation was confirmed to facilitate ACE2-binding [58] and now roots for the observed packing enhancement in the presence of L452R.

### 3.2. FEP Calculations Revealed Synergy between Mutations Enhancing RBD-ACE2 Binding

By simple decomposition of interaction energies between RBD and ACE2 molecules in MD simulations, we may conclude that both electrostatic and van der Waals (vdW) interactions contribute to the enhanced binding caused by the mutations (Appendix A). However, only a limited conclusion could be drawn from these data due to large fluctuations and exclusion of contributions from explicit water molecules in the calculation. To quantitatively evaluate the effects of mutation sets found on the RBD towards ACE2-binding, we also subjected the equilibrated structures of WT RBD-ACE2 complex to alchemical free energy calculations. The free energy perturbation (FEP) method, with a well-designed thermodynamics cycle (Appendix A), was widely used to measure binding affinity changes due to point mutations (alchemical alterations) and proven to be in excellent agreement with experimental binding assays for many biomolecular systems [32,34,60]. To ensure the reproducibility of such binding affinity calculations and to obtain enough configurational samplings for mutated binding interfaces between the molecules, we performed an order of magnitude more sets of individual calculations than that typically used in previous studies (Appendix A). We measured an enhancement in the binding free energy of association upon both sets of mutations. The RBD_mut1_ was shown to enhance binding by −2.27 ± 0.27 kcal/mol, while that of RBD_mut2_ was −6.22 ± 0.28 (Figure 2C and Table 1). These results agree with the net increase in buried areas observed above for both mutated binary complexes.

Interestingly, based on estimated macromolecular binding efficiency [61], the enhancement of binding originated from the increased buried areas (roughly 50–60 Å^2^) would be 3–4 kcal/mol. To account for the discrepancy in binding affinities, we also separately performed FEP calculations for individual single mutations (Figure 2C and Appendix A and Table 1). Previous experiments indicated that a single mutation L452K or L452R alone could only enhance RBD-ACE2 binding by approximately 2–6-folds [10,62], which corresponds to a <1 kcal/mol enhancement and agrees well with our binding affinity calculation results of −0.87 ± 0.29 kcal/mol (Figure 2C and Table 1). Then we sought to compare the dissimilar component in the mutation sets. Mutations E484Q and, especially, T478K reside in the CR1 region (at the N-terminal end) of the RBD-ACE2 interface which is dominated by hydrophobic contacts between Phe486 of RBD and fencing residues Leu79, Met82 and Tyr83 of ACE2. Both mutations approximately take place at a distance on two sides of the hydrophobic pocket and therefore were expected to only marginally enhance the affinity to ACE2 [63,64]. As a result, both mutations E484Q and T478K reside at the periphery of a hydrophobic contact region. However, as indicated by our FEP calculations, much stronger bindings (i.e., more negative ∆∆G than the simple addition of two individual mutants, or synergies) were found for both double mutations with E484Q or T478K on top of L452R (Figure 2C and Table 1). Interestingly, unlike Thr478, Glu484 electrostatically interacts with Lys31 of ACE2 by forming salt-bridges (Appendix A), and therefore its mutation to Gln would presumably disrupt the local electrostatic complementarity, leading to a less strong binding as one could anticipate from a purely structural perspective. Therefore, where is the synergy coming from? More intriguingly, T478K is spatially far away from any RBD-ACE2 contact region. Thus, again where is the synergy coming from or how is this potentially allosteric enhancement in binding achieved?

Taking T478K with L452R (RBD_mut2_) as an example, we first suspected that the synergetic enhancement in binding affinity comes from simultaneously mutating two residues into positively charged ones (L452R + T478K) which greatly increases the surface electrostatics of the RBD and might attract more strongly the ACE2 to bind. However, no significant decrease in spatial distance between the residue 478 of RBD and the ACE2 was found in our MD simulations of RBD_mut2_ (Appendix A). We then decomposed the electrostatic and vdW contributions to the synergetic enhancement using an exponential approach. We found that the vdW interactions clearly dominate during mutations to RBD_mut2_, consistent with the surged buried surface area observed in the MD simulations (Appendix A). We then analyzed our MD simulation trajectories to uncover the origin of the synergetic effect between distant single mutations from the viewpoint of protein structural dynamics of the RBD. Both Pearson cross-correlation (Appendix A) and the generalized correlation (Figure 3A–C), which captures both linear and non-linear correlated motions of residues based on Shannon entropy [65], revealed that only the L452R + T478K mutation set significantly strengthens the correlations between residue 452 and residues 469–492 (magenta box in Figure 3C), spanning over the whole CR1 region of RBD (Figure 3C below). The coupled motions, through communication between distant sites, were found to be mediated through a hydrophobic cage formed by residues Thr470, Il472, Phe490, Pro491, and Leu492 (Figure 4A). Consistently, in all our simulations (including the WT), we observed a high motion correlation between residue 452 (Leu or Arg) and Gln493 (black box in Figure 3A–C), in agreement with how we anticipated in the last section that mutations to Leu452 would facilitate the engagement of Gn493 in enhanced bindings to ACE2.

However, as both RBD_mut1_ and RBD_mut2_ largely retained their interaction network with ACE2 (Figure 1B), it is challenging to reveal the exact conformational shifts which directly enhance ACE2-binding due to coupled motions. In turn, we looked at the changes in hydration level near the mutated residues (Appendix A) and found a surprisingly large set of stable hydrogen-bonding or water-mediated polar interactions (i.e., almost half of the FEP simulations, considering only the end states) form between residues Glu484 and Gln493 of the RBD. Water-mediated polar interactions are defined when a pair of residues is connected by one (Figure 4B) or two (Figure 4C) water molecules in the network of hydrogen bonds and have been shown to play important structural roles in other systems [66] (see the Appendix A for details). Interestingly, we found that although the water-mediated hydrogen bonds exist in individual mutations (L452R and T478K), none of them show a dramatic increase as observed in the L452R + T478K mutation set (Figure 4D). We therefore anticipate that the coupled motions progressively promote water-mediated polar interactions between Glu484 and Gln493, which in turn creates enhanced protein structural stability in the region and finally leads to enhanced ACE2-binding without altering the interaction network at the interface. Consistently, such an intricate and responsive hydrogen-bonding network does not exist in RBD_mut1_ as Glu484 is mutated to Gln, explaining the less synergy between these two individual mutations. Similar enhancement in ACE2-binding by reshaping the binding interface or altering structure of RBD due to mutated residues has been reported in previous studies [36,67], but has not been studied in light of correlated motions between individual mutated residues.

Furthermore, as we shall also show in a later section, these mutations would also drive the emergence and spread of the VOC lineages by lowering the effectiveness of neutralizing monoclonal and/or polyclonal antibodies [12]. This allosteric effect preserves its dominance in other binding simulations between RBD and mAbs, as demonstrated in the following section.

### 3.3. Mutated RBDs Show Immunoevasion to Several mAbs

To this end, we have demonstrated that rigorous computational chemistry tools could provide critical insights into how mutated RBDs manipulate their binding with ACE2. On the other hand, VOCs have also shown antibody evasion in both computational and experimental studies [64]. To gain an understanding of the structural origin that governs RBD-mAbs binding and the affinity changes due to point mutations, we also performed a comprehensive set of FEP calculations (Appendix A) by choosing mAbs with distinct modes of actions. In contrast to the classification proposed by Barnes et al. [26], we categorized RBD-mAbs complexes according to the binding epitopes on RBD determined by binding poses (Figure 5A) [68]. One critical difference is between the monoclonal antibodies COVA2-39 and P2B-2F6, which were classified together into “Class 2” by Barnes and coworkers but are now distinguished by binding poses 2 and 3 in our current characterization. In other words, the defined binding poses of mAbs are all based on their unique and distinct binding regions of RBM. Again, we performed alchemical mutagenesis studies on both RBD_mut1_ and RBD_mut2_ (L452R + E484Q and L452R + T478K), in which the binary complexes were subjected to at least 200 ns unbiased MD simulations before FEP calculations (see the Appendix A for details).

As already illustrated for the RBD–ACE2 binding interface, the L452R mutation does not directly hinder binding of antibodies competing with ACE2 (e.g., Pose 1 mAbs). Both RBD_mut1_ and RBD_mut2_ have shown a negligible change in binding affinity for mAb B38 (Figure 5B and Table 1). In fact, none of the mutations we studied here participates directly in the interactions with neutralizing antibodies binding through Pose 1 (Figure 5C). Consistently, mAb BD-629 was also found to effectively neutralize the viruses containing the L452R mutation in experiments [69] and bind through a similar manner compared to B38.

COVA2-39 (Pose 2 mAb) forms a stable polar interaction network in CR1 region, with Gly54 (backbone only) and Thr56 (backbone and sidechain) constantly forming hydrogen bonds with Glu484 of RBD (Figure 5D and Appendix A). In accordance, mutating Glu484 to Gln severely disrupted the electrostatic interactions and thus weakened the binding (Appendix A). Therefore, we observed a BFE of 4.53 ± 0.59 kcal/mol for L452R + E484Q and 4.92 ± 0.48 kcal/mol for E484Q (a high energetic penalty for this single mutation) (Figure 5B and Table 1). COVA2-39 does not bind to the far-outer parts of CR1 where Thr478 locates, neither does it pose to the CR2 region so that Leu452 is also away from the interaction interface (Figure 5D). Consistently, no contribution from L452R and T478K alone was observed (Appendix A). L452R + T478K combines two non-contributing mutations, raising almost no impact to binding affinity (0.52 ± 0.33 kcal/mol). In addition, the complementarity-determining region loop of COVA2-39 containing Gly54 and Thr56 sits in-between Gln493 and Glu484 of RBD (Appendix A), further weakening the water-mediated interaction network between Glu484 and Gln493 in RBD_mut2_.

P2B-2F6 (Pose 3 mAb) was found to also have significant energetic penalty due to disruption of hydrophobic contacts in the CR2 region, leading to weakened binding in both RBD_mut1_ and RBD_mut2_ (Figure 5B and Table 1). In CR2 region, Ile103 and Val105 on the heavy chain formed a hydrophobic pocket accommodating Leu452 in WT RBD (Figure 5E). Introducing a charged Arg easily disrupts the local hydrophobic interactions. Glu484 of RBD was found to only occasionally form hydrogen bonds with sidechain of Lys55 on the light chain of P2B-2F6 (Appendix A). Accordingly, we observed a substantial loss in BFE for single L452R mutation (7.46 ± 0.57 kcal/mol) and almost no difference for single E484Q mutation (0.75 ± 0.34 kcal/mol) (Table 1 and Appendix A). Similar conclusion has been made through experiments, in which a comprehensive escape map revealed that binding of LY-CoV555 could easily be abolished by mutations at L452 alone [70,71]. Together, RBD_mut1_ achieved weakened binding by a significant 8.35 ± 0.75 kcal/mol, which is comparable to that from the single mutation L452R.

What is interesting about Pose 3 mAbs was again the synergy between L452R and T478K mutations. Similar to the case of Pose 2 mAbs, Thr478 does not directly participate in the interactions with the antibody (Figure 5E), confirmed by a BFE change of −0.61 ± 0.34 kcal/mol for single T478K mutation (Table 1 and Appendix A). As one would anticipate, RBD_mut2_ (L452R + T478K) will reflect the contribution from individual single mutations, mainly from L452R in this case. However, this mutation set surprisingly show a less weakened binding of 4.25 ± 1.03 kcal/mol. We found that although Pose 3 mAbs could also sterically block interactions between Gln493 and Glu484 like Pose 2 mAbs, the water-mediated polar network still presents due to the slight shift in binding angle of the antibody (Appendix A). More interestingly, hydrogen-bonding analysis show that Glu484 in RBD_mut2_ forms stable interactions with sidechain of Arg111 and backbone of Ala108 of P2B-2F6 (Appendix A), instead of the Lys55 as described for WT. As a result, RBD_mut2_ showed a less weakened binding due to newly formed hydrogen bonds. This observation again emphasized the importance of the accurate characterization of binding modes of mAbs to the RBD.

To confirm our results of combinative mutants evading Pose 3 mAbs, we also performed BFE calculations on a bivalently bound structure (Pose1 + 3), in which two mAbs simultaneously bind to RBD through Pose 1 and Pose 3 respectively. For this structure, we obtained a free energy loss of 6.92 ± 1.57 kcal/mol for RBD_mut1_ and 4.47 ± 0.91 kcal/mol for RBD_mut2_ (Figure 5B and Table 1), indicating that interaction with the Pose 3 mAb was disrupted while that with the Pose 1 mAb was barely impaired.

Lastly, Pose 4 mAbs such as CR3022 does not directly bind to RBM (Figure 5A) and served as a control for our BFE calculations. As expected, our results showed that bindings of Pose 4 mAbs are slightly favored for most of the mutations we studied here (Table 1 and Appendix A), indicating that the free state estimations are biased within 1 kcal/mol. The only exception is a large enhancement for RBD_mut1_ (Appendix A) which might indicate a systematic deviation of ~2 kcal/mol in the free state estimations of this mutation set. Difficulties in the accurate estimation of free energy changes for specific mutations will be discussed in the following section.

## 4. Discussion

SARS-CoV-2 has undergone a delicate evolution process from SARS-CoV to become a highly transmissible virus. Owing to the substantial efforts from both genomic and structural perspectives, we have gathered solid understanding of how SARS-CoV-2 is different from its ancestral strain SARS-CoV in terms of the RBM and other Spike domains [58]. It comes as no surprise when scientists discovered that most of the amino acid mutations to RBD are deleterious for ACE2 binding [63,72]. It is fully foreseeable that the virus will progressively reach its form of maximum transmission without further significant advances in binding ability towards the host. On the other hand, due to increasing recovery and vaccination rate, the emergence of variants with novel antigenic profiles threatening to diminish the neutralization of antibodies is also expected. The fact that RBM or peripheral RBM modifications were still observed in emerging new variants, especially the VOCs, implies that these mutations not only could retain the binding affinity to ACE2 but also present extra advantages to the survival of the virus, probably by means of immune evasion. As fully illustrated by the Omicron variant, a full explanation for the gain in overall infectiousness of newly emerged VOCs would require more than simply a higher RBD-ACE2 binding affinity.

As shown by our simulation results, interplays between seemingly independent mutations simultaneously enhance and impair macromolecular interactions in order to become highly evasive to mAbs while remained highly infectious. Our findings therefore provide important molecular insights into emerging antigenic shift of SARS-CoV-2. Intriguingly, Gheeraert et al. who combined MD simulations with perturbation contact network analysis also revealed a synergistic long-range effect from L452R and T478K altering the hydrophobic cluster around the Cys480-Cys488 disulfide bridge (i.e., the CR1 region) as well as the distant ACE-binding interface. The limitations of our quantitative results are also obvious: although alchemical free energy calculations were shown to reproduce experimentally derived differences in the binding affinities of biomolecules [73,74], the results in general contained much larger errors especially when dealing with charge-changing mutations. In addition, it is possible that the preferred protonation state changes for some titratable residues upon binding. Nevertheless, our large sets of replicas were consistent in showing a non-additivity effect in the binding affinities of the concerned mutations. However, detection of such effect is challenging as the Delta variant was reported to be only three-fold more transmissible and < 10-fold more resistant to convalescent plasma when compared to previous strains [4,23,75], even though we observed significant increases in RBD–ACE2 interaction interface area and enhanced remote hydrogen-bond network from multiple dynamics trajectories. Additionally, we carried out reversed mutations in which a Delta RBD was mutated back to wild-type and observed a less severe synergetic effect (~1 kcal/mol) along this alchemical path, mainly due to uncertainties in calculations for the free state (i.e., the spike protein alone) (Appendix A). Noteworthy, the interquartile range of most calculations spread over ~2 kcal/mol which is twice the margin of error for most of the computed free energy differences (~1 kcal/mol assuming 98% confidence interval) (Appendix A). This again emphasizes that computing an accurate magnitude of the observed nonaddictive term is challenging for FEP calculations. The focus of this work is to uncover, from a molecular simulation perspective, such a synergetic effect between two seemingly unrelated single mutations found in a viral sublineage strain which caused the second major global wave of COVID-19.

The fact that there is extensive person-to-person variation in how mutations affect plasma antibody binding and neutralization prompts the need to focus on avidity in addition to affinity of the RBD-mAbs binding events. For instance, the neutralizing activity of several plasma samples could be reduced by >10-fold by a single mutation while a few samples were essentially unaffected [12]. This also poses a limitation to our simulation studies as the selected mAbs may not fully reflect the immune responses in individuals. Nevertheless, dynamics trajectories from MD simulations provide knowledges of complex molecular interactions which are transferrable to a much larger set of mAbs through rapidly advancing structural characterization techniques [76]. We would also anticipate that recovering thermodynamics of the neutralization process concerning accumulative Spike mutation sets through rigorous computational chemistry approaches will draw increasing attention as more structural data of Spike complexes become available over time.

## 5. Conclusions

In this study, we performed MD simulations and FEP calculations concerning two combinative sets of mutations of RBD—L452R + E484Q (Kappa) and L452R + T478K (Delta) to address the molecular mechanism behind the dramatic increase in infectiousness of B.1.617 lineage of SARS-CoV-2. Our alchemical estimation of the RBD-ACE2 binding free energy changes for Kappa is −2.27 ± 0.27 kcal/mol, mainly due to a net increase in buried molecular surface between the proteins. Although no direct interaction between Thr478 of RBD and ACE2 was observed, the Delta yielded a stronger binding of −6.22 ± 0.28 kcal/mol. By analyzing the dynamic trajectories, we uncovered an unexpected water-mediated interaction network between Gln493 and Glu484 of RBD, which is allosterically influenced by the spatially distant mutation pair (L452R + T478K) and contributes substantially to the molecular interactions on the surface of the domain. We further evaluated the binding free energy changes towards several mAbs classified by their binding poses to RBD. Both mutation sets were shown to effectively evade mAbs that directly bind RBD at its central hydrophobic region containing Leu452 or the charged patch containing Glu484. We discovered that disruption to the binding of mAbs we studied here was driven by direct removal of hydrophobic or electrostatic interactions. Interestingly, the distant mutation (T478K) was found to slightly facilitate binding of the antibody through coupled residual motions at Gln493, but overall remained highly evasive when combined with L452R. Accordingly, molecular mechanisms behind acquisition of mutations at Gln493 in several Omicron subtypes would require further investigation. Taken together, we demonstrated the feasibility of in silico benchmarking combinative mutation sets on the increased transmissibility and level of neutralizing antibody evasion. Interplays between individuals within accumulative mutation sets open up a new direction to evaluate the transmissibility of not only newly emerged SARS-CoV-2 strains, but also future pathogenic coronaviruses.

## Figures and Tables

**Figure 1 biomolecules-12-01742-f001:**
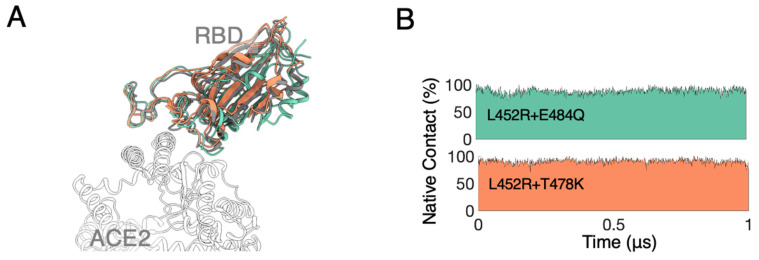
(**A**) ACE2 (white) in complex with WT RBD (grey), RBD_mut1_ (L452R + E484Q and indigo) and RBD_mut2_ (L452R + T478K and orange). Complexes were aligned according to ACE2. (**B**) Native contacts between mutated RBDs and ACE2, taking the WT complex as the reference.

**Figure 2 biomolecules-12-01742-f002:**
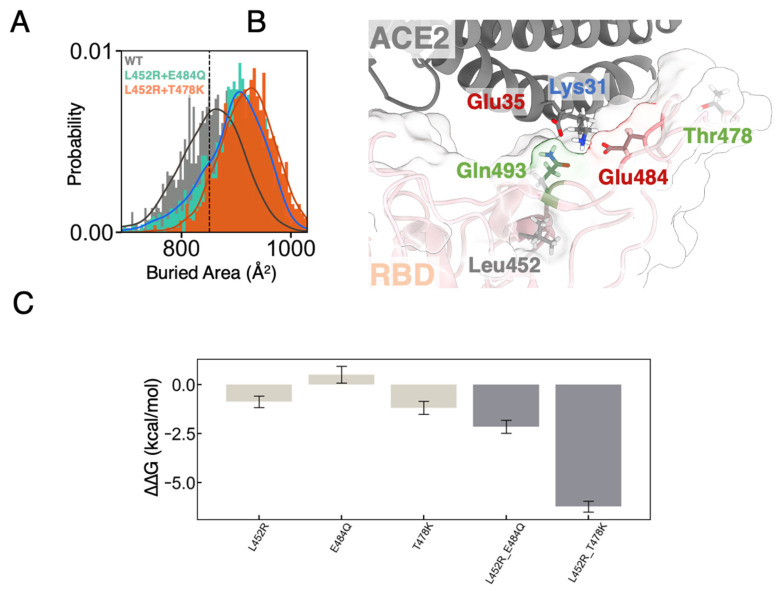
(**A**) Buried area between RBD and ACE2 measured from MD simulations. Dotted line indicates measurement for the crystal structure (PDB: 6M0J). (**B**) Residual interaction network at the binding interface between RBD and ACE2. ACE2 was shown as grey cartoon and RBD was shown as pink cartoon with white transparent molecular surface. (**C**) Change in BFE of mutated RBDs for ACE2. Error bars were the standard errors.

**Figure 3 biomolecules-12-01742-f003:**
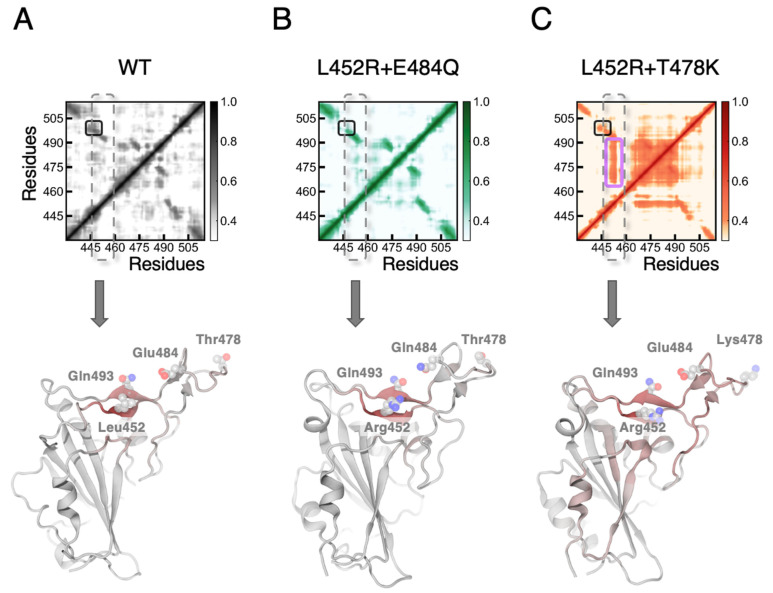
Generalized correlation analysis for wild type (**A**), mutation L452R + E484Q (**B**), and mutation L452R + T478K (**C**). Above are the generalized correlation matrices. Black and magenta boxes indicate the correlations between residue 452 and residue 493 or residues 469–493, respectively. Grey dashed boxes indicate the correlation involved residues 450 to 454. Below are mapping of averaged correlation with residues 450 to 454 onto every other residue. Red colors represent the degrees of correlation while white colors indicate no correlation. Key residues are shown as spheres.

**Figure 4 biomolecules-12-01742-f004:**
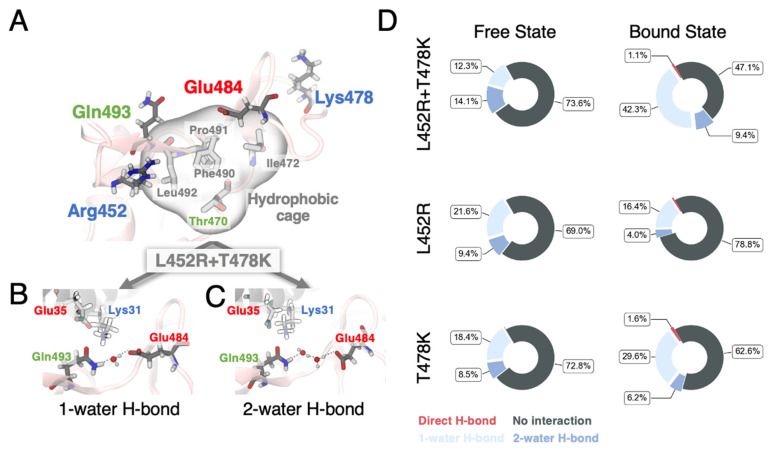
(**A**) Residual interaction network on the surface of RBD_mut2_ (L452R + T478K). Key residues were shown as sticks. The molecular surface of the hydrophobic cage was shown in transparent. (**B**,**C**) Snapshot from MD simulations showing hydrogen bond network between Gln493 and Glu484 mediated through one (**B**) or two (**C**) water molecule(s). (**D**) Statistics of polar interactions (direct or water-mediated hydrogen bonds) between Gln493 and Glu484 from FEP calculations.

**Figure 5 biomolecules-12-01742-f005:**
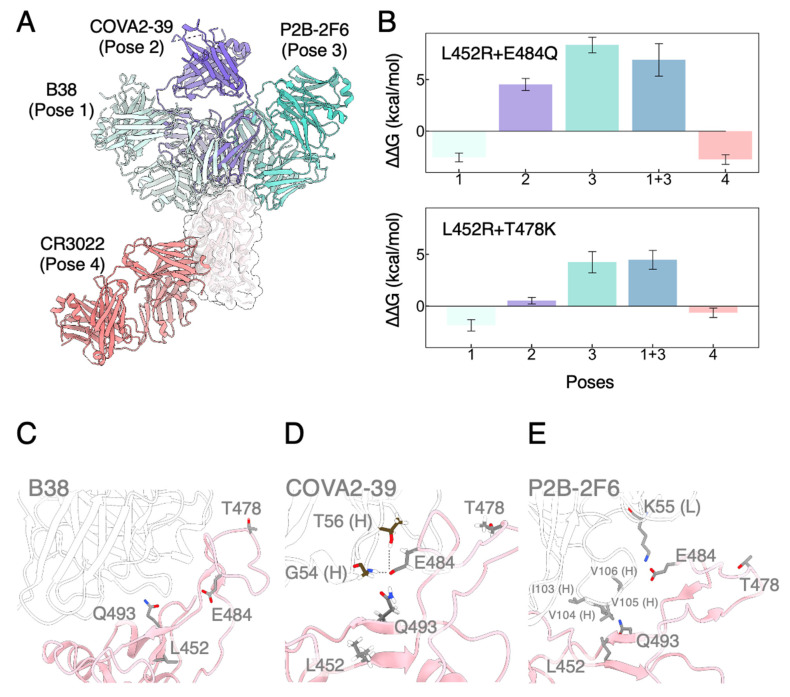
(**A**) RBD (white) in complex with mAb of different binding poses. (**B**) Change in BFE of RBD_mut1_ (L452R + E484Q) and RBD_mut2_ (L452R + T478K) for mAbs. Error bars were the standard errors. (**C**–**E**) Residual interaction network at the mAb-binding interfaces of (**C**) B38 (Pose 1), (**D**) COVA2-39 (Pose 2) and (**E**) P2B-2F6 (Pose 3). RBD and the mAb were colored in pink and white respectively. Key residues were shown as sticks.

**Table 1 biomolecules-12-01742-t001:** Change in BFE for ACE2 and mAbs bound through different poses.

		L452R + E484Q	L452R + T478K	L452R	E484Q	T478K
ACE2		−2.27 ± 0.27	−6.22 ± 0.28	−0.87 ± 0.29	0.51 ± 0.42	−1.18 ± 0.33
mAbs	Pose 1	−2.54 ± 0.42	−1.86 ± 0.55	−1.29 ± 0.30	−1.68 ± 0.26	−0.71 ± 0.43
	Pose 2	4.53 ± 0.59	0.52 ± 0.33	−1.04 ± 0.24	4.92 ± 0.48	1.38 ± 0.38
	Pose 3	8.35 ± 0.75	4.25 ± 1.03	7.46 ± 0.57	0.75 ± 0.34	−0.61 ± 0.34
	Pose 1 + 3	6.92 ± 1.57	4.47 ± 0.91			
	Pose 4	−2.75 ± 0.47	−0.65 ± 0.46	−0.95 ± 0.24	−0.74 ± 0.38	−0.88 ± 0.41

All free energies in kcal/mol.

## Data Availability

PDB structures and sequence alignment data were downloaded from https://www.rcsb.org/. The data presented in this study are available upon request from the corresponding author and also archived on Zenodo.

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
