# Peer review of "SARS-CoV-2 Delta Variant: Interplay between Individual Mutations and Their Allosteric Synergy"

_biomolecules, 2022, doi:10.3390/biom12121742_

Round 1

Reviewer 1 Report

This ms demonstrates that 26 new variants of SARS-CoV-2 accomplish both “attack” (infection) and “defense” (antibody neutral- 27 ization escape) with the same “polished sword” (mutated Spike RBD). This is an important study to tackle a challenge problem of mutationsthe of SARS-CoV-2 Spike RBD. Authors used molecular dynamics simulations, in combination with free energy perturbation calculations, to examine the effects of two combinative mutation sets, L452R + E484Q and L452R + T478K. These dy- namic trajectories show an enhancement in binding affinity between mutated RBD and the com- 19 mon receptor protein angiotensin converting enzyme 2 (ACE2) through a net increase in buried 20 molecular surface area of the binary complex.

This reviewer highly recommend this ms should be publishable.

Author Response

We sincerely thank the reviewer for this positive feedback. 

Reviewer 2 Report

To Authors: The authors have executed the molecular dynamics simulations of the SARS-CoV-2 S-glycoprotein of delta variant in complex with ACE2 receptor. The authors proposed the synergy between L452R and T478K mutation sites significantly contributed the affinity of the proteins. Although the simulations were executed properly, and the results were presented appropriately, this reviewer is very much concerned that their results were not sufficiently compared with many more similar simulation studies of delta variant S-glycoprotein. This reviewer cannot be positive to recommend this article for publication unless this point is sufficiently fixed.

1)      A lot of molecular dynamics studies of delta-variant S-protein and ACE2 or antibodies have published recently, for example, Socher et al. CSBJ 20, 1168 (2022), Hwang et al., J. Chem. Inf. Model. 62, 1771 (2022), Zhou et al., J. Chem. Inf. Model. 62, 4512 (2022), Pipitò et al., BioEssays. 44:2200060 (2022), Gheeraert et al., J. Chem. Inf. Model. 62, 3107 (2022), Mahmood et al., Immun Inflamm Dis. 10:e683 (2022), and Cheng et al., iScience 25, 103939 (2022). Many of these reports examined antibody complexes, and some of them detected synergy between mutation sites. Therefore, this reviewer could not be positive to recommend this report for publication unless the presented results were sufficiently compared with the published results. It would be the responsibility of the authors of this study to demonstrate what part of their result could add something new to the knowledge.

2)      The authors used the systems which contained single RBD domain of S-protein. S-glycoprotein is the homo-trimer of subunits consisting of more than 1200 amino acid residues and highly glycosylated. Some of the published MD studies were executed on a whole (glycosylated) trimer system, and seemingly it would produce much more realistic results, even though the major interactions are those between RBD and ACE2. This reviewer think that some rationalizations should be provided for justifying the usage of the smaller system.

3) The Materials and Methods section in the main text is concisely written. However, at least the source of coordinates and composition of the simulation systems should be explicitly written in the main text.

Reviewer 3 Report

I congratulate the authors on an extremely thorough investigation of the synergistic interplay between the Delta variant mutations L452R and T478K in their dual roles of enhanced binding to ACE2 and immune evasion from mAbs, specifically those binding close to the region where these mutations are localized (“Pose-3” mAbs). 

(Optional suggestion):

It’d be very beneficial to the readers, if the authors provided suggestions for what they think are residues (or 3-4 residue motifs) in interacting therapeutic molecules like mAbs or nanobodies that can counteract the immune evasion observed in Pose-3 mAbs. These can serve as heuristics in mAb design against these particular mutations should they reappear as the major functional mutations in other future VoCs.

Author Response

We sincerely thank the reviewer for this positive feedback. We also thank the reviewer for asking for further insights into design of therapeutic molecules (mAbs or nanobodies) based on the immune evasion originated from the observed interplays between mutations. As the reviewer has also pointed out, the mutations we studied have already reappeared as the major functional mutations in some future VoCs. Design of “Pose-3” therapeutic molecules which specifically target these particular mutations will be presented in a future study. 

Round 2

Reviewer 2 Report

To Authors: The authors have sufficiently responded to the comments of this reviewer, except for the one mentioned in the following comment.

Authors’ reply 1: “There have been a number of published MD studies on differential bindings between SARS-CoV-2 mutants and either ACE2 or antibodies. … but from a slightly different perspective [J. Chem. Inf. Model. 62, 3107 (2022)]. “ What the authors are required is not to explain to this reviewer but to the readers. This reply should be written (in a little concise manner) in the main text. This reviewer recommends this manuscript for publication after the authors fix this point.
